# Exploring Anthocyanin and Free and Bound Phenolic Compounds from Two Morphotypes of Araçá (*Psidium cattleianum* Sabine) by LC-ESI-QqQ-MS/MS

**DOI:** 10.3390/foods12173230

**Published:** 2023-08-28

**Authors:** Patrícia Gotardo Machado, Danielle Santos Londero, Milene Teixeira Barcia, Cristiano Augusto Ballus

**Affiliations:** 1Department of Food Science and Technology, Federal University of Santa Maria (UFSM), Santa Maria 97105-900, Brazil; patricia.machado@acad.ufsm.br (P.G.M.); milene.barcia@ufsm.br (M.T.B.); 2Health Sciences Center, Federal University of Santa Maria (UFSM), Santa Maria 97105-900, Brazil; londero.danielle@acad.ufsm.br

**Keywords:** bioactive compounds, native fruits, LC-MS/MS, antioxidant capacity, chemometrics, correlation

## Abstract

Araçá is a Brazilian native fruit belonging to the Myrtaceae family. Although some studies already prove its health benefits, it is still necessary to explore the phenolic compounds in all its parts separately. This study aimed to investigate the free, esterified, glycosylated, and insoluble phenolics in two morphotypes of araçá, red and yellow, evaluating peel, pulp, and seed separately, using liquid chromatography coupled to mass spectrometry (LC-ESI-QqQ-MS/MS). Fourteen phenolics and five anthocyanins were quantified in both morphotypes. The peels presented the highest contents, followed by the pulp and seeds. Red araçá stood out over the yellow one only in the phenolic fractions resulting from the peel, with the yellow araçá being superior in the phenolic fractions of the pulp and seed. The highest antioxidant capacities were detected for the pulp-esterified phenolics (498.3 µmol g^−1^) and peel-free phenolics (446.7 µmol g^−1^) of yellow araçá. Principal component analysis (PCA) indicated specific markers to differentiate the samples. All parts of the araçá proved to be a rich source of phenolic compounds, in different fractions, mainly in the peel. This information will be beneficial to stimulate the consumption of native fruits and their possible use in the development of new products.

## 1. Introduction

Brazil has favorable geographical and climatic conditions for fruit production, being the world’s third-largest producer of oranges, bananas, pineapples, papayas, grapes, and apples [1]. In addition, Brazilian biodiversity presents a range of native fruit trees, many of which are not widely known even by the scientific community [2]. Native fruits have received more attention because of their antioxidant properties, demonstrating that these fruits can be sources of vitamin C, flavonoids, and phenolic acids [3,4]. The nutritional appeal of a healthy diet is growing, and the intake of bioactive compounds decreases the probability of developing chronic diseases, such as cancer, Alzheimer’s, and cardiovascular and pulmonary diseases [5]. According to Teixeira et al. [6], only 25% of Brazilian fruits’ alleged health benefits have been proven through studies. Plants found in the Brazilian flora have contributed to developing therapeutic alternatives by identifying their specialized metabolites. Botanical species belonging to the Myrtaceae family, mainly of the genus *Psidium* L., are used to prevent diseases and symptoms associated with several body systems [7].

The araçá (*Psidium cattleianum* Sabine) is a native Brazilian fruit belonging to the Myrtaceae family and can be found in several states of the country, as well as in Uruguay. Its fruits are small, in the form of a berry, measuring an average of 2 cm in diameter and with numerous seeds, with a yellow or red epicarp when ripe [8].

Fruits with red epicarp and endocarp are the rarest, with 80% having cream and white endocarp [9]. According to Beltrame et al. [10], red fruits have a higher content of phenolic compounds.

Some previous studies on the phenolic compounds in red and yellow araçás exist. Pereira et al. [11] investigated the total phenolics by the Folin–Ciocâlteu method and antioxidant capacity by the DPPH method. They identified the phenolic compounds using HPLC-QTOF-MS in the red and yellow morphotypes but only in the peel and pulp of the fruits. Medina et al. [12] evaluated only the pulp of red and yellow fruits, analyzing the total anthocyanins and carotenoids using the UV/Vis spectrophotometric method, antioxidant capacity by DPPH, total phenolics by the Folin–Ciocâlteu method, and quantification of phenolic compounds by HPLC. Mallmann et al. [13] quantified and identified the free and bound phenolic compounds by LC-DAD-ESI-MS/MS of whole red and yellow araçá fruits and performed ORAC analysis.

According to Medina et al. [12], araçá, in addition to its pleasant flavor, also has an anticancer action, where the authors found that fruit extracts reduced the survival rates of breast cancer and colon cancer cells, in addition to having a high content of vitamin C and antioxidant capacity. In their study, Dos Santos Pereira et al. [9] found that araçá contains chemical compounds of interest, such as minerals, fatty acids, sugars, volatile compounds, and carotenoids. Although there are already some studies proving the benefits of the fruit, it is still necessary to complete some gaps in the knowledge about it, such as knowing the phenolic compounds in all its parts separately.

Unlike the other cited studies, our work separately analyzed phenolic compounds in all parts of the fruit, peel, pulp, and seed. The objective of the present study was to extract and quantify the free, esterified, glycosylated, and insoluble phenolic compounds from red and yellow araçá fruits, evaluating them separately in the peel, pulp, and seed. In addition, the anthocyanins present in the peel and pulp of the red araçá were also extracted and quantified.

## 2. Materials and Methods

### 2.1. Chemicals and Reagents

All reagents and chemicals used were HPLC, LC-MS, or analytical (≥98% purity). They were: gallic acid (≥99%), protocatechuic acid (≥98), 4-hydroxybenzoic acid (≥99%), syringic acid (≥98%), ferulic acid (≥99%), vanillic acid (≥97%), chlorogenic acid (≥98%), *p*-coumaric acid (≥98%), ellagic acid (≥98%), caffeic acid (≥97%), catechin (≥98%), epicatechin (>97%), quercetin (≥97%), quercetin-3-glucoside (≥90%), taxifolin (≥95%), myricetin (≥97%), pelargonidin-3-glucoside (≥98%), cyanidin-3-glucoside (≥95%), delphinidin-3-glucoside (≥95%), malvidin-3-glucoside (≥90%), and peonidin-3-glucoside (≥95%). They were purchased from Sigma-Aldrich (St. Louis, MO, USA) and Carbosynth (BioSynth, Staad, Switzerland). Ultrapure water was obtained from a purification system (Millipore, Bedford, MA, USA).

### 2.2. Collection and Preparation of Samples

The two morphotypes of araçá (*Psidium cattleianum* Sabine) (Figure 1) were obtained during the fruit harvest season, from mid-September 2021 to January 2022, in the municipality of Santo Augusto (27°51′03″ south latitude, 53°46′38″ longitude west, and 528 m altitude), where they were collected in different locations within the city.

Both trees were mainly isolated and did not have interference concerning solar irradiation. The harvest was carried out manually, encompassing different positions of the trees (upper, lower, and central) to guarantee greater sample homogeneity. Sequentially, the suitable fruits were sanitized, pulped, and separated into three parts, peel, pulp, and seeds. The fruit parts were frozen for 24 h (−18 °C), lyophilized for 48 h, crushed in a knife mill, packaged, and stored (−18 °C) until the beginning of the extractions.

### 2.3. Extraction of Phenolic Compounds

The extraction of phenolic compounds occurred according to Arruda et al. [14], where 1 g of lyophilized samples of araçá (pulp, peel, or seed) was weighed and extracted with 15 mL of methanol:acetone:water (7:7:6, *v*/*v*/*v*) solution. The samples were ultrasonicated in an ultrasound bath (power of 640 Watts and a frequency of 40 kHz) for 30 min at room temperature and centrifuged at 3500× *g* for 5 min. Then, the supernatants were collected, and the residues were extracted thrice under the same conditions. Soon after, the supernatants were combined and used to fractionate soluble phenolic compounds, and the solid residue was stored to determine insoluble phenolic compounds.

#### 2.3.1. Extraction of Phenolic Fractions

As detailed in the following sections, free, esterified, glycosylated, and insoluble phenolic compounds were obtained according to Arruda et al. [14]. The entire experimental process of this study is summarized in Figure 2.

#### 2.3.2. Free Phenolic Compounds

The supernatants obtained in the previous step (2.3) were evaporated under vacuum at 35 °C to remove the organic solvents, and sequentially the aqueous phase was acidified to pH 2.0 using HCl (6 mol L^−1^) and centrifuged at 3600× *g*, for 5 min. Supernatants were extracted thrice with 15 mL of hexane to remove interfering lipids. Free phenolic compounds were extracted thrice with 10 mL of diethyl ether: ethyl acetate (1:1, *v*/*v*). The organic phases were combined and filtered using anhydrous sodium sulfate and filter paper evaporating to dryness under vacuum at 35 °C. The dried residues, with the fraction of free phenolic compounds, were dissolved in 5 mL of HPLC-grade methanol and filtered through a 0.22 µm PTFE membrane. The aqueous phases were stored to continue the extractions.

#### 2.3.3. Esterified Phenolic Compounds

The aqueous phase remaining after the extraction of free phenolics was hydrolyzed with NaOH (4 mol L^−1^) containing EDTA (10 mM) and 1% ascorbic acid (1:2, *v*/*v*) for 4 h at room temperature. The pH of the hydrolyzate was adjusted to 2.0 using HCl (6 mol L^−1^), and the phenolic compounds released from the soluble esters were extracted three times, as previously described in item Section 2.3.2.

#### 2.3.4. Glycosylated Phenolic Compounds

The remaining aqueous phase from the extraction of esterified phenolic compounds was hydrolyzed with 5 mL of HCl (6 mol L^−1^) for 60 min at 75 °C. Soon after, they were extracted three times, as described in item Section 2.3.2.

#### 2.3.5. Insoluble Phenolic Compounds

The solid residue, which was obtained after extracting the total phenolic compounds, was hydrolyzed with NaOH (4 mol L^−1^) containing 10 mM EDTA and 1% ascorbic acid in a solid-to-solvent ratio of 1:20 (*w*/*v*) for 4 h at room temperature. The pH was adjusted to 2.0 using HCl (6 mol L^−1^) and centrifuged at 3600× *g* for 5 min. The supernatant was extracted with 15 mL of hexane three times to remove interfering lipids. The insoluble phenolic compounds were extracted three times, as described in item Section 2.3.2.

### 2.4. Extraction of Anthocyanins

Anthocyanins were exhaustively extracted from the peel and pulp of red araçá, following Neves et al. [15]. Aliquots of 1 g of the lyophilized samples were extracted with 25 mL of the water:methanol:formic acid solution (48.5:50:1.5 *v*/*v*/*v*) using ultrasound (40 kHz, 640 W) for 5 min. Then, the samples were centrifuged for 10 min at 3600× *g*. The same procedure was performed four more times, and, in the end, the supernatants were pooled and stored. The extraction efficiency was evaluated using liquid chromatography coupled to mass spectrometry (LC-ESI-QqQ-MS/MS), whose conditions are described in Section 2.6. After five extractions, a yield of 99.4% of anthocyanins was obtained in the samples. Sequentially, the extracts were purified to remove any interferents present. For purification, SPE cartridges type C_18_ were used, according to the methodology of Farias et al. [15].

### 2.5. Quantification of Phenolic Compounds by LC-ESI-QqQ-MS/MS

Free, esterified, glycosylated, and insoluble phenolic compounds were separated and quantified by liquid chromatography with a diode array detector coupled to a triple quadrupole mass spectrometer (LCMS 8045, Shimadzu, Kyoto, Japan) containing an electrospray ionization source (ESI) and triple quadrupole *m*/*z* analyzer. The compounds were separated according to Machado et al. [16].

A Zorbax RRHD Eclipse C-18 (XDB) reverse phase column (Agilent, Santa Clara, CA, USA) was used, with 2.1 mm × 150 mm and 1.8 µm particle size and a flow rate of 0.2 mL min^−1^, oven temperature of 35 °C, and injection volume of 10 μL. For the mobile phases, acidified water (0.5% acetic acid) was used as mobile phase A and acetonitrile as mobile phase B. The chromatographic gradient was formed by mobile phase B of 0.01 min, 0%; 0.01–20 min, 20%; 20–21 min, 27%; 21–23 min 30%; 23–30 min, 30%; 30–40 min, 50%; 40–50 min, 75%; 50–60 min, 100%; 60–63 min, 0%; 63–70 min, 0%, totaling 70 min of chromatographic run.

The ESI source was operated in negative mode, using nebulizing gas flow 2 L min^−1^, drying gas flow 4 L min^−1^, heating gas flow 6 L min^−1^, interface temperature 350 °C, interface voltage −3.5 kV, desolvation line temperature 150 °C, and heat block temperature 200 °C. The *m*/*z* values for all ions and other multiple reaction monitoring (MRM) parameters can be found in Table 1.

### 2.6. Quantification of Anthocyanins by LC-ESI-QqQ-MS/MS

Anthocyanins were also separated and quantified by LC-ESI-QqQ-MS (LCMS 8045, Shimadzu, Kyoto, Japan). The separation method was defined according to Farias et al. [15]. For the separation, a Zorbax Eclipse XDB C-18 reverse phase column (Agilent, Santa Clara, CA, USA), 150 mm, 2.1 mm, 3.5 μm particle size, with a flow rate of 0.19 mL min^−1^ was used, with an oven temperature of 40 °C and 10 µL of injection volume. The solvents used were water, acetonitrile, methanol, and formic acid (88.5:3:8:0.5 *v*/*v*/*v*) as mobile phase A, and water, acetonitrile, methanol, and formic acid (41.5:50:8:0.5 *v*/*v*/*v*) as mobile phase B.

Mobile phase B chromatographic gradient was 0–8 min, 3%; 8–28 min, 30%; 28–34 min, 50%; 34–38 min, 100%; 38–40 min, 100%; 40–46 min, 3%; 46–48 min, 3%, totaling 48 min of chromatographic run. The ESI source was operated in positive mode, with the parameters of nebulizing gas flow 2.0 L min^−1^, drying gas flow 4.0 L min^−1^, heating gas flow 6.0 L min^−1^, interface voltage 4.58 kV, interface temperature 346 °C, desolvation line temperature 163 °C, and heating block temperature 200 °C.

Individual standards of each anthocyanin were used for quantification by external calibration curve, except for petunidin-3-glucoside, which was quantified with the delphinidin-3-glucoside curve. The two product ions with the highest signal were used via multiple reaction monitoring (MRM) to quantify and confirm the compounds. All *m*/*z* values for ions and other parameters are available in Table 1.

### 2.7. Methods Validation

The LC-ESI-QqQ-MS method (Table 1) was validated according to Thompson et al. [17]. Analytical curves were constructed with ten equidistant points in triplicate and prepared in random order. Statistical tests for outlier detection, Levene’s test, and ANOVA were performed on the results obtained. To determine the limits of detection (LOD) and quantification (LOQ), the angular coefficients of the calibration curves and the standard deviation of the area of the lowest point of the curve were used.

### 2.8. Antioxidant Capacity by ORAC (Oxygen Radical Absorbance Capacity)

The ability to deactivate the peroxyl radical was performed according to Ou et al. [18], modified by Huang et al. [19] and Dávalos et al. [20], for all extracts obtained from free, esterified, glycosylated, and insoluble phenolic fractions. Polystyrene microplates with 96 wells, specific for fluorescence tests, were used. The reading was performed in a microplate reader, for 80 min, at wavelengths of 485 nm (excitation) and 520 nm (emission).

### 2.9. Statistical Analysis

The Shapiro–Wilk test was used to verify normality and the Levene test to ascertain the homoscedasticity of the data. Sequentially, analysis of variance (ANOVA) and Tukey’s test were used to verify statistical differences between samples, with a significance level (*p* ≤ 0.05). Pearson’s correlation coefficients between phenolic compounds of both fruits and their antioxidant capacities were calculated using RStudio software version 2023.03.1 (Rstudio, Boston, MA, USA), with graphs generated using the ggcorrplot package. Principal component analysis (PCA) was applied to all samples (2 samples, in 3 parts—peel, pulp, and seed—all in triplicate, totaling 18 samples), and considering all compounds that were detected in the different fractions plus the antioxidant capacity, totaling 56 variables. Thus, a matrix of 18 samples × 56 variables was generated for the PCA. Before the PCA, the data were auto-scaled. PCA was performed using Pirouette 3.11 software (Infometrix, Bothell, WA, USA).

## 3. Results and Discussion

### 3.1. Methods Validation

The liquid chromatography–mass spectrometry methods were validated to guarantee reliable results (Table 1). The values obtained for *r*^2^ were above 0.99, and the data were average and homoscedastic. The limits for detection and quantification were low and adequate for the expected compound contents.

### 3.2. Quantification of Anthocyanins

Anthocyanins are responsible for the coloring of fruits; they also increase their resistance to damage from ultraviolet rays and attack by pathogens. They can be used as substitutes for artificial colors [21,22]. The red araçá, due to its color, indicates the potential presence of anthocyanins to be studied.

Cyanidin-3-glucoside is the anthocyanin most commonly distributed in fruits, mainly in red and magenta colors [23]. According to Figure 3, it is the major anthocyanin present in the peel and pulp of the fruit, followed by petunidin-3-galactoside. It is still possible to observe in the peel the presence of peonidin-3-glucoside, pelargonidin-3-glucoside, and malvidin-3-glucoside; however, in the pulp, only pelargonidin-3-glucoside is present, albeit discreetly.

Vinholes et al. [24] evaluated the content of total monomeric anthocyanins in the red araçá fruit as 29.3 mg cyanidin-3-glucoside equivalent per 100 g of fresh fruit. Previously, Medina et al. [12] investigated the levels of total monomeric anthocyanins only in the fruit pulp, where they obtained a concentration of 6.29 mg of cyanidin-3-glucoside in 100 g of the fruit. In our study, the peel and pulp of the fruit were evaluated separately. For the pulp, values of 0.70 mg g^−1^ and the pulp of 0.15 mg g^−1^ of cyanidin-3-glucoside were found.

### 3.3. Quantification of Phenolic Compounds

So far, the literature does not present studies on the characterization and quantification of the phenolic compounds present in red and yellow araçá when evaluated separately in peel, pulp, and seed and of the free, esterified, glycosylated, and insoluble phenolic fractions. The results obtained from the characterization and quantification of these compounds are shown in Table 2.

Fourteen phenolic compounds were quantified, five of them flavonoids (catechin, epicatechin, myricetin, quercetin-3-glucoside, and taxifolin) and nine phenolic acids (gallic, protocatechuic, 4-hydroxybenzoic, caffeic, vanillic, syringic, ferulic, *p*-coumaric, and ellagic). All compounds were found in both araçá species.

Regarding the phenolic fractions, the fruits of araçá differ among themselves. For the peel, the free fraction (67.94%) of the red araçá stood out over the other phenolic fractions, while in the yellow araçá, the insoluble fraction (25.54%) was predominant in the fruit.

The same was observed for pulps. In the red araçá, the phenolic fraction in more significant proportion was the glycosylated one (8.66%), whereas for the yellow araçá, it was the esterified fraction (13.87%). However, in the seeds, it was observed that in both fruits, the glycosylated fraction was higher than the others, being 82.23% for the red araçá and 80.09% for the yellow araçá.

Considering the results obtained for both fruits, it is possible to observe that the red araçá stood out over the yellow one only in the phenolic fractions resulting from the peel, with the yellow araçá being superior in the phenolic fractions of the pulp and seed.

In both fruits, the major compounds quantified were catechin and gallic acid, the first being observed in higher concentrations in yellow araçá and the second in red araçá.

Catechins are considered safe and easy to apply phytochemicals in foods [25]. They are present in high concentrations in various fruits, vegetables, and herbal beverages, and even though they are not essential for human nutrition, they help prevent several diseases [26].

In our work, it was verified that the concentrations of catechin in yellow araçá were 1571.5 µg g^−1^ in the pulp, 1353.1 µg g^−1^ in the pulp, and 69.9 µg g^−1^ in the seed. According to Pereira et al. [11], catechin was found in more significant proportions in the pulp and seed of yellow araçá 2.07 and 189 µg g^−1^, respectively. Still, in line with the results found in this study, Mallmann et al. [13] observed that in the whole araçá fruit, catechin was found in higher concentrations in the yellow fruit 229.3 µg g^−1^ than in the red fruit 168.7 µg g^−1^.

It is also possible to observe that catechin is available in greater concentration in the free phenolic fractions, indicating that it was not linked to other matrix components.

Catechins are the main precursors of condensed tannins. When they unite, they are also known as procyanidins and are most commonly found in fruit peels and seeds [27].

Gallic acid is a phenolic compound widely found in fruits and vegetables. It is a promising compound for developing new drugs, as it has antioxidant, anticancer, antibacterial, antifungal, antiviral, anti-inflammatory, and antidiabetic activities [28,29]. It is the second primary compound found in both araçás. For yellow araçá, the esterified and insoluble phenolic fractions were the highest for both parts of the fruit. With the red araçá, the esterified and insoluble phenolic fractions were observed in a more significant proportion for peel and seed. For the fruit pulp, the esterified and glycosylated fractions were superior.

Non-extractable phenolic compounds remain in the sample residue from previous extractions and may be released through hydrolysis treatments [13]. The gallic acid was bound to other matrix components. During the acid and alkaline hydrolysis, the connections of the gallic acid with the matrix were broken, allowing it to be obtained in free form.

Medina et al. [12] evaluated only the yellow and red araçá pulps with two extractions and found that gallic acid values were between 279.4 and 726.7 µg g^−1^ in the yellow araçá pulp and 193.2 and 801.0 µg g^−1^ in the red araçá pulp. All parts of the fruit were evaluated in our work, and higher results than those in the cited study were obtained. For the yellow araçá, 999.5 µg g^−1^ was found and for the red one, 9508.6 µg g^−1^.

### 3.4. Antioxidant Capacity by Oxygen Radical Absorbance Capacity (ORAC)

The antioxidant capacity through the ORAC method for the free, esterified, glycosylated, and insoluble phenolic fractions was evaluated for the first time in red and yellow araçás fruits separately in their parts, peel, pulp, and seed.

As observed in the quantification of phenolic compounds, the antioxidant capacity values also differed between the two araçá species (*p* < 0.05). As shown in Figure 4, the peel extracts of both fruits have a higher antioxidant capacity, followed by the pulp and seeds sequentially.

Analyzing the peels, it was possible to observe that the esterified phenolic fraction (498.3 µmol g^−1^) was the highest for the yellow araçá, followed by the insoluble (387.5 µmol g^−1^), glycosylated (233.8 µmol g^−1^), and in lower free concentration (226.6 µmol g^−1^), while for red araçá, the fraction with the highest concentration was free (409.2 µmol g^−1^), followed by insoluble (236.4 µmol g^−1^), glycosylated (217.7 µmol g^−1^), and esterified (201.8 µmol g^−1^).

Concerning the pulps, it was observed that the yellow araçá indicated a greater antioxidant capacity in the free fraction (446.7 µmol g^−1^). For the other fractions, the results were close, with the glycosylated fraction representing (177.8 µmol g^−1^), esterified (174.2 µmol g^−1^), and insoluble (165.7 µmol g^−1^). In the case of red araçá pulp, the esterified fraction (397.4 µmol g^−1^) was the highest, followed by the free (270.8 µmol g^−1^), insoluble (215.4 µmol g^−1^), and glycosylated (210.2 µmol g^−1^).

For the fruit seeds, in the yellow genotype, it was observed that the insoluble fraction (191.9 µmol g^−1^) stood out over the others, the glycosylated one (166.5 µmol g^−1^), and the esterified fractions (137.5 µmol g^−1^) and free (130.2 µmol g^−1^) are close. It appears that the seeds of the red genotype have the glycosylated fraction (205.2 µmol g^−1^) in evidence, sequentially the insoluble fractions (176.2 µmol g^−1^), esterified (109.5 µmol g^−1^), and free (101.2 µmol g^−1^).

Notably, the esterified, glycosylated, and insoluble fractions influence the antioxidant capacity of the fruits, which reinforces the importance of carrying out acid and alkaline hydrolysis since they are responsible for breaking the bonds of the compounds with the matrices, facilitating their extraction.

### 3.5. Correlation between Phenolic Compounds and Antioxidant Capacity

Pearson’s correlation coefficients between phenolic compounds and antioxidant capacities in fruit peel, pulp, and seeds are shown in Figure 5 (red araçá) and Figure 6 (yellow araçá). So far, this statistical evaluation of the correlation between araçá and its fractions has not been found in the literature.

The peel of the red araçá has the highest correlation of phenolic compounds with the antioxidant capacity, followed by the fruit’s pulp and seeds (Figure 5). Thirteen compounds positively correlated with the antioxidant capacity were found in the bark, namely: cyanidin-3-glucoside > pelargonidin-3-glucoside > protocatechuic acid > peonidin-3-glucoside > petunidin-3-glucoside > vanillic acid > gallic acid > malvidin-3-glucoside > epicatechin > catechin > quercetin-3-glucoside.

It is possible to observe that, although catechin and gallic acid are the major compounds quantified in the fruit fraction, they have a low correlation with the antioxidant capacity. On the other hand, cyanidin-3-glucoside, consistent with the quantification, is the major anthocyanin in the fruit peel and has the highest correlation with the antioxidant capacity.

For the red araçá pulp, it was found that seven phenolic compounds have a positive correlation with the antioxidant capacity, presented in the following order: cyanidin-3-glucoside > pelargonidin-3-glucoside > epicatechin > syringic acid > quercetin-3-glucoside > 4-hydroxybenzoic acid > *p*-coumaric. It can be seen that, as in the fruit peel, cyanidin-3-glucoside and pelargonidin-3-glucoside have the highest correlations with antioxidant capacity.

Seeds showed the lowest positive correlations, being only among the four compounds caffeic acid > 4-hydroxybenzoic acid > myricetin > *p*-coumaric acid.

Unlike the red araçá, the yellow genotype showed the highest correlation with the antioxidant capacity in the pulp, followed by the seeds and peel. The pulp showed a positive correlation between six phenolic compounds and antioxidant capacity, namely vanillic acid > 4-hydroxybenzoic acid > protocatechuic acid > quercetin-3-glucoside> catechin > caffeic acid.

Yellow araçá seeds stood out, compared to red araçá, with a correlation with the antioxidant capacity of five phenolic compounds: syringic acid > 4-hydroxybenzoic acid > catechin > myricetin > quercetin-3-glucoside. Differing from the red genotype, the yellow peel had the lowest correlation between phenolic compounds and antioxidant capacity, with only three compounds, ellagic acid > 4-hydroxybenzoic acid > caffeic acid, presenting a positive correlation.

### 3.6. Principal Component Analysis (PCA)

Principal component analysis (PCA) was used to assess whether some of the analyzed compounds could be considered markers to classify and differentiate the studied samples. Figure 7a,b presents the results as scores and loadings graphs. It was possible to explain 60.04% of the data variance using two principal components.

From Figure 7a, it can be seen that the peel of the red araçá was separated from the peel of the yellow araçá. Although expected, we can now determine which analyzed compounds are essential to generate this differentiation. From Figure 7b, we understand that the red araçá peel is positively correlated with the significant presence of anthocyanins, in addition to ellagic and protocatechuic acids from the free fraction; 4-hydroxybenzoic acid from the glycosylated fraction; vanillic and ferulic acids from the insoluble fraction; and *p*-coumaric acid from the esterified fraction. The yellow araçá peel is positively correlated with high ORAC values in the esterified, insoluble, and glycosylated fractions; in addition to ellagic and *p*-coumaric acids from the insoluble fraction; epicatechin from the esterified and insoluble fractions; quercetin3-glucoside from the insoluble fraction; and 4-hydroxybenzoic acid from the esterified fraction.

On the other hand, the red and yellow araçá pulps do not present very marked differences concerning the analyzed compounds. Even with the absence of anthocyanins in the yellow araçá pulp, both pulps were located very close in the scores graph (Figure 7a), positively correlating with the presence of high levels of caffeic and ferulic acids in the esterified fraction and vanillic acid in the free fraction (Figure 7b).

The red and yellow araçá seeds were located at a certain distance from each other (Figure 7a), even though it was not as large as observed for the peels. The myricetin of the esterified fraction mainly distinguishes the seed of the red araçá. In contrast, the seed of the yellow araçá has a positive correlation with high levels of ellagic acid in the glycosylated fraction (Figure 7b).

Thus, from different compounds in different fractions, it was possible to classify the peel, pulp, and seed of the araçá samples separately. Within the samples of peels and seeds, it was possible to differentiate the yellow and red morphotypes of the araçá.

## 4. Conclusions

The free, esterified, glycosylated, and insoluble phenolic compounds were quantified for the first time in red and yellow araçá and their parts (peel, pulp, and seed). LC-ESI-QqQ-MS/MS quantified fourteen phenolic compounds, the two significant compounds being catechin and gallic acid, in both fruits and their parts. Five anthocyanins were also quantified, and cyanidin-3-glucoside was the majority in the peel and pulp of red araçá, followed by petunidin-3-galactoside.

It was observed that the araçá fruits differ among themselves in all the results obtained. The red araçá stood out over the yellow one only in the phenolic fractions resulting from the skin, with the yellow araçá being superior in the phenolic fractions of the pulp and seed.

Regarding the antioxidant capacity of the fruits, the esterified, glycosylated, and insoluble phenolic fractions were superior to the free fraction, reinforcing the importance of separating the phenolic compounds during extraction. In addition, both fruits have a positive correlation between the quantified phenolic compounds and the antioxidant capacity, and the red araçá showed better results than the yellow one.

Applying the principal component analysis, it was possible to determine some markers among the analyzed compounds that respond by classifying the samples separately.

From the information obtained in our study, we have a detailed evaluation of the composition of phenolic compounds, in their different fractions, in all constituent parts of red and yellow araçá. This study helps to fill a gap in the knowledge of phenolics in the peel, pulp, and seeds of araçá, as well as providing data that can be used to stimulate the consumption of these fruits and their study as raw material for the development of new products.

## Figures and Tables

**Figure 1 foods-12-03230-f001:**
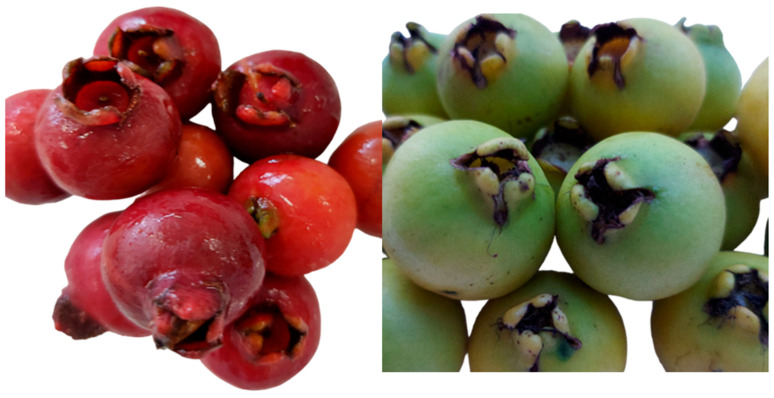
Samples of red and yellow araçá.

**Figure 2 foods-12-03230-f002:**
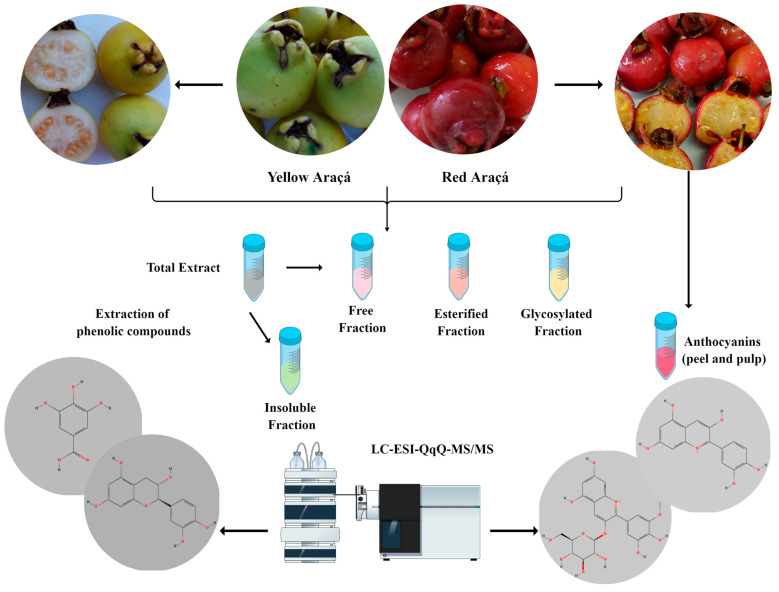
The flowchart summarizes the experimental process’s main steps for obtaining anthocyanin and non-anthocyanin phenolic compounds from the peel, pulp, and red and yellow araçá seed.

**Figure 3 foods-12-03230-f003:**
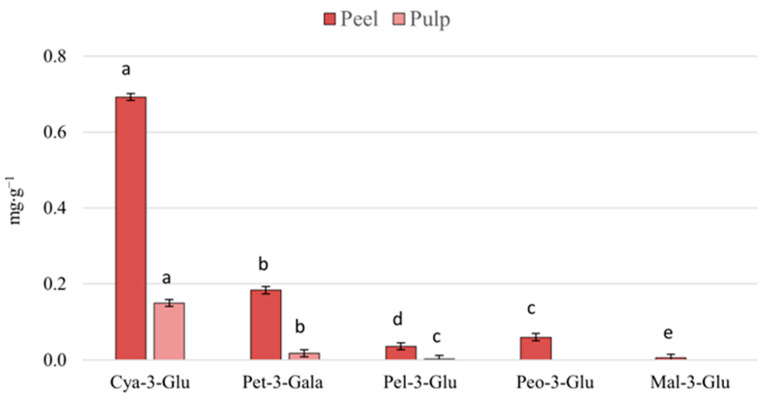
Contents of anthocyanins present in the peel and pulp of red araçá. Cya-3-Glu: Cyanidin-3-Glucoside; Pet-3-Gala: Petunidin-3-Galactoside; Pel-3-Glu: Pelargonidin-3-Glucoside; Peo-3-Glu: Peonidin-3-Glucoside; Mal-3-Glu: Malvidin-3-Glucoside. Means followed by the same letter in the column do not differ, according to the Tukey test (*p* ≤ 0.05).

**Figure 4 foods-12-03230-f004:**
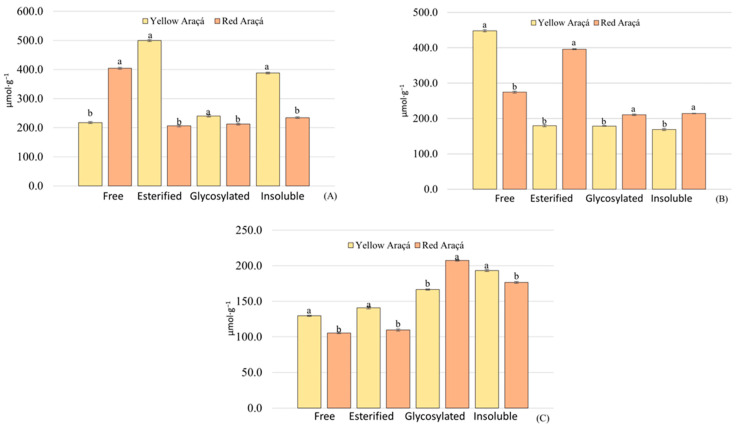
Antioxidant capacity of yellow araçá and red araçá. Pulp (**A**), peel (**B**), and seed (**C**). Means followed by the same letter in the column do not differ, according to the Tukey test (*p* ≤ 0.05).

**Figure 5 foods-12-03230-f005:**
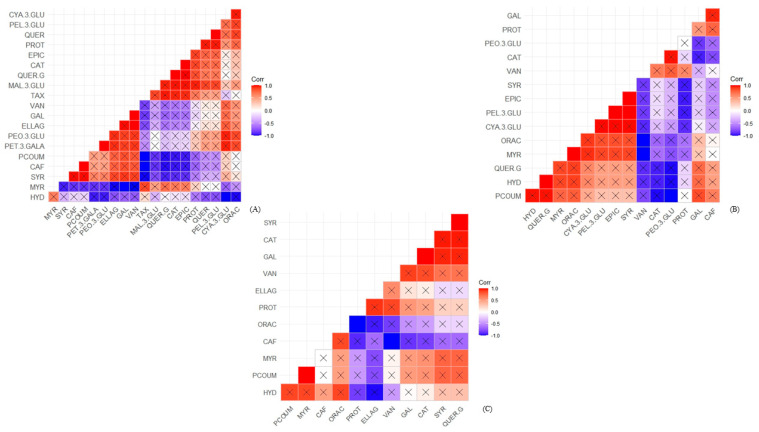
Pearson correlation analysis between phenolic compounds and antioxidant capacity in red araçá. Pulp (**A**), peel (**B**), and seed (**C**). Quadrants marked with an “X” correspond to statistically significant correlations. CYA.3.GLU: Cyanidin-3-Glucoside; PEL.3.GLU: Pelargonidin-3-Glucoside; QUER: Quercetin; PROT: Protocatechuic acid; EPIC: Epicatechin; CAT: Catechin; QUER.G: Quercetin-3-Glycoside; MAL.3.GLU: Malvidin-3-Glucoside; TAX: Taxifolin; VAN: Vanillic acid; GAL: Gallic acid; ELLAG: Ellagic acid; PEO.3.GLU: Peonidin-3-Glucoside; PET.3.GALA: Petunidin-3-Galactoside; PCOUM: *p*-Coumaric acid; CAF: Caffeic acid; SYR: Syringic acid; MYR: Myricetin; HYD: 4-Hydroxybenzoic acid.

**Figure 6 foods-12-03230-f006:**
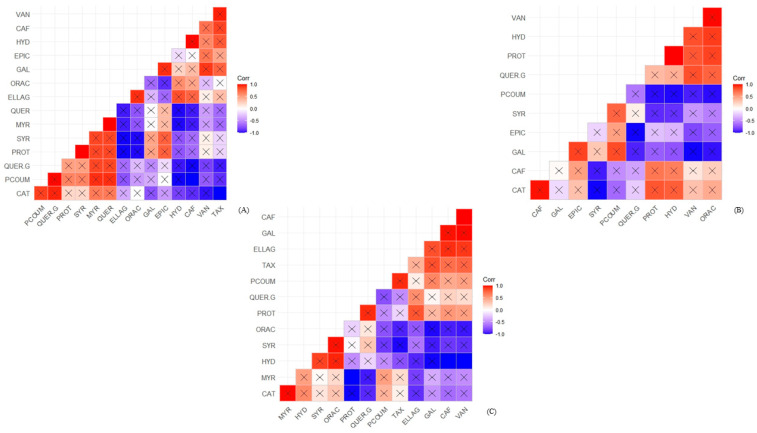
Pearson correlation analysis between phenolic compounds and antioxidant capacity in yellow araçá. Pulp (**A**), peel (**B**), and seed (**C**). Quadrants marked with an “X” correspond to statistically significant correlations. VAN: Vanillic acid; CAF: Caffeic acid; HYD: 4-Hydroxybenzoic acid; EPIC: Epicatechin; GAL: Gallic acid; ELLAG: Ellagic acid; QUER: Quercetin; MYR: Myricetin; SYR: Syringic acid; PROT: Protocatechuic acid; QUER.G: Quercetin-3-Glycoside; PCOUM: *p*-Coumaric acid; CAT: Catechin; TAX: Taxifolin.

**Figure 7 foods-12-03230-f007:**
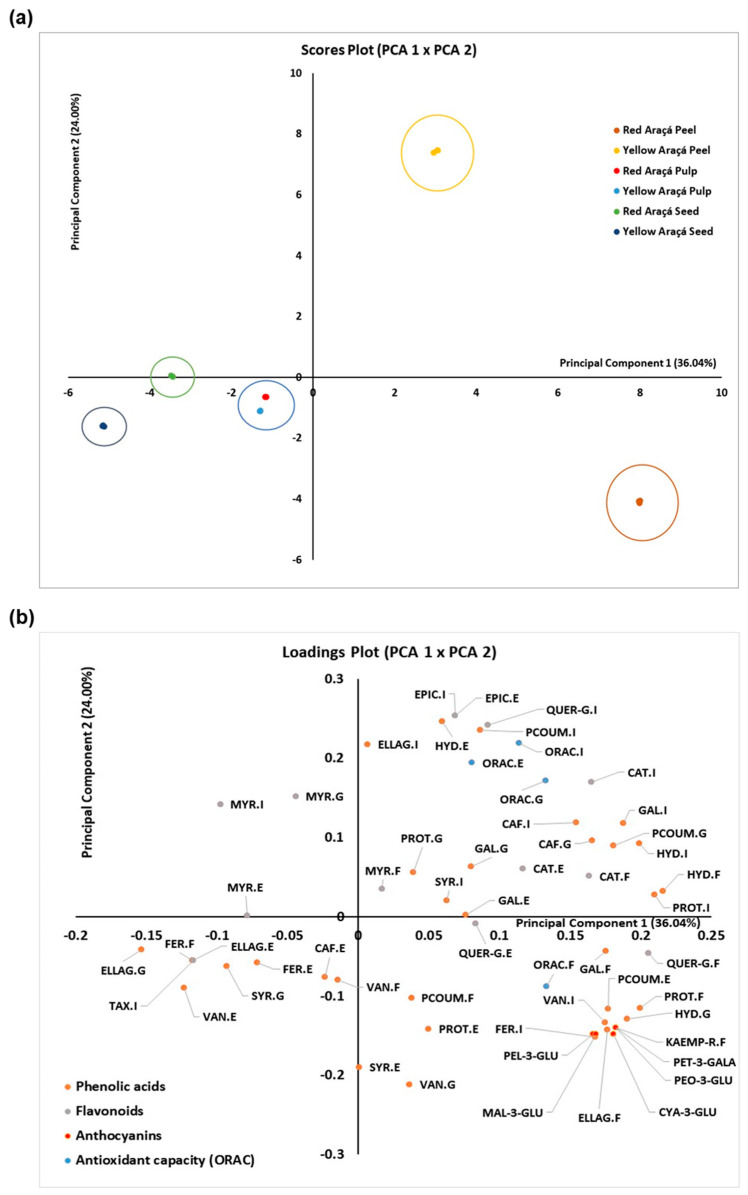
PCA results, considering the first and second principal components (PC1 × PC2). (**a**) Scores plot. (**b**) Loadings plot. Abbreviations: CYA-3-GLU: Cyanidin-3-Glucoside; PEL-3-GLU: Pelargonidin-3-Glucoside; QUER: Quercetin; PROT: Protocatechuic acid; EPIC: Epicatechin; CAT: Catechin; QUER-G: Quercetin-3-Glycoside; MAL-3-GLU: Malvidin-3-Glucoside; TAX: Taxifolin; VAN: Vanillic acid; GAL: Gallic acid; ELLAG: Ellagic acid; PEO-3-GLU: Peonidin-3-Glucoside; PET-3-GALA: Petunidin-3-Galactoside; PCOUM: *p*-Coumaric acid; CAF: Caffeic acid; SYR: Syringic acid; MYR: Myricetin; HYD: 4-Hydroxybenzoic acid. The letters F, G, E, and I after the abbreviation of each compound indicate the fraction to which it belongs (free, glycosylated, esterified, or insoluble, respectively).

**Table 1 foods-12-03230-t001:** Experimental parameters for multiple reaction monitoring (MRM) transitions and analytical parameters used for method validation.

Compounds	Transitions (*m*/*z*)	Retention Time (min)	Q1 (V) ^a^	Q3 (V) ^a^	CE (V) ^a^	LOD ^a^ (mg L^−1^)	LOQ ^a^ (mg L^−1^)	*r* ^2^	Lack of Fit Test (*p* > 0.05)
Gallic acid	169.20–125.10 * 169.20–78.90 **	6.92	14.0 14.0	23.0 15.0	16.0 22.0	0.0202	0.0061	0.9976	0.7231
Protocatechuic acid	153.10–109.05 153.10–108.00	9.65	13.0 13.0	12.0 19.0	15.0 23.0	0.0002	0.0006	0.9902	0.9218
4-Hydroxybenzoic acid	137.20–93.00 137.20–65.05	12.34	29.0 29.0	16.0 24.0	16.0 30.0	0.0007	0.0023	0.9919	0.1188
Chlorogenic acid	353.10–191.20 353.10–84.90	12.07	17.0 29.0	21.0 14.0	17.0 46.0	0.0032	0.0053	0.9926	0.0871
Catechin	289.20–245.10 289.20–108.95	12.24	14.0 14.0	26.0 12.0	15.0 26.0	0.0014	0.0023	0.9934	0.0629
Caffeic acid	179.10–135.15 179.10–134.05	13.53	13.0 14.0	14.0 24.0	16.0 26.0	0.0011	0.0016	0.9920	0.0971
Vanillic acid	167.15–152.00 167.15–108.05	13.65	15.0 13.0	27.0 12.0	15.0 19.0	0.0013	0.0046	0.9983	0.0971
Epicatechin	289.10–245.15 289.10–109.00	13.74	15.0 13.0	18.0 21.0	14.0 23.0	0.0032	0.0019	0.9942	0.060
Syringic acid	197.00–182.15 197.00–123.05	14.07	18.0 14.0	19.0 24.0	14.0 24.0	0.0011	0.0013	0.9928	0.1474
*p*-Coumaric acid	163.10–119.10 163.10–93.05	16.24	15.0 15.0	13.0 18.0	15.0 32.0	0.0003	0.0017	0.9944	0.0905
Ferulic acid	193.00–134.05 193.00–178.15	18.57	14.0 14.0	14.0 19.0	17.0 14.0	0.0001	0.0002	0.9931	0.8921
Ellagic Acid	300.90–284.00 300.90–145.15	20.22	20.0 20.0	29.0 25.0	30.0 37.0	0.0007	0.0009	0.9928	0.2409
Taxifolin	303.10–285.15 303.10–125.05	20.55	14.0 15.0	21.0 13.0	12.0 21.0	0.0011	0.0036	0.9926	0.6177
Quercetin-3-Glycoside	463.20–300.00 463.20–301.10	21.14	17.0 13.0	30.0 20.0	27.0 23.0	0.0002	0.0005	0.9912	0.5951
Myricetin	317.10−151.10	24.30	15.0	28.0	24.0	0.00024	0.00062	0.9905	0.7309
	317.10−179.10		15.0	19.0	20.0				
Pelargonidin-3-Glucoside	432.90–271.00 432.90–121.00	17.83	−11.0 −11.0	−26.0 −18.0	−22.0 −55.0	0.00013	0.00076	0.9922	0.1477
Cyanidin-3-Glucoside	448.90–286.95 448.90–137.00	15.36	−15.0 −12.0	−27.0 −20.0	−24.0 −55.0	0.00022	0.000694	0.9919	0.1530
Malvidin-3-Glucoside	493.00–331.00 493.00–314.95	20.70	−13.0 −10.0	−20.0 −30.0	−22.0 −50.0	0.0100	0.0390	0.9967	0.0746
Delphinidin-3-Glucoside	464.90–303.00 464.90–229.00	11.01	−12.0 −12.0	−29.0 −21.0	−23.0 −55.0	0.0022	0.0067	0.9913	0.2792
Peonidin-3-Glucoside	463.00–301.00 463.00–286.05	19.46	−16.0 −16.0	−29.0 −27.0	−21.0 −43.0	0.0122	0.0361	0.9949	0.2954

* First transition used for quantitation; ** second transition used for identification. ^a^ Q1 and ^a^ Q3: pre-bias; ^a^ CE: collision energy; V: voltage. LOD: limit of detection; LOQ: limit of quantification.

**Table 2 foods-12-03230-t002:** Quantification of free, esterified, glycosylated, and insoluble phenolic compounds present in the peel, pulp, and seed of red araçá and yellow araçá (mean ± standard deviation, n = 3).

Compounds	Araçá Samples	Peel (µg g^−1^)				Pulp (µg g^−1^)				Seed (µg g^−1^)			
		F *	E *	G *	I *	F *	E *	G *	I *	F *	E *	G *	I *
Catechin	Red	1040.4 ^a^ (±0.3)	151.6 ^b^ (±0.07)	nq	202.6 ^b^ (±0.5)	158.7 ^b^ (±0.1)	81.0 ^b^ (±0.1)	nq	14.3 ^a^ (±0.1)	nq	90.5 ^a^ (±0.1)	nq	nq
	Yellow	1034.6 ^a^ (±0.1)	174.1 ^a^ (±0.5)	nq	362.8 ^a^ (±0.3)	1105.3 ^a^ (±0.7)	235.2 ^a^ (±0.2)	nq	12.6 ^b^ (±0.1)	60.672 ^a^ (±0.1)	nq	nq	nq
Epicatechin	Red	nq	nq	nq	nq	nq	nq	nq	nq	nq	nq	nq	nq
	Yellow	nq	9.6 ^a^ (±0.1)	nq	nq	nq	nq	nq	2.6 ^a^ (±0.7)	nq	nq	nq	nq
Myricetin	Red	7.5 ^a^ (±0.1)	nq	nq	nq	nq	nq	8.533 ^a^ (±0.4)	nq	8.1 ^b^ (±0.1)	8.4 ^a^ (±0.1)	nq	9.4 ^a^ (±0.1)
	Yellow	7.5 ^a^ (±0.1)	nq	9.6 ^a^ (±0.7)	8.1 ^a^ (±0.1)	nq	nq	nq	nq	9.2 ^a^ (±0.2)	nq	8.2 ^a^ (±0.1)	8.0 ^b^ (±0.2)
Quercetin-3-Glycoside	Red	31.0 ^a^ (±0.5)	6.1 ^a^ (±0.1)	nq	5.7 ^b^ (±0.1)	10.6 ^b^ (±0.3)	nq	nq	nq	8.2 ^a^ (±0.4)	4.4 ^a^ (±0.5)	nq	nq
	Yellow	16.1 ^b^ (±0.5)	4.9 ^b^ (±0.1)	nq	49.1 ^a^ (±0.8)	18.4 ^a^ (±0.2)	4.9 ^a^ (±0.1)	nq	nq	nq	4.5 ^a^ (±0.1)	nq	nq
Taxifolin	Red	nq	nq	nq	nq	nq	nq	nq	nq	nq	nq	nq	nq
	Yellow	nq	nq	nq	nq	nq	nq	nq	nq	nq	nq	nq	7.3 ^a^ (±0.1)
Gallic acid	Red	173.5 ^a^ (±0.4)	1260.5 ^a^ (±0.1)	341.4 ^b^ (±0.3)	919.7 ^a^ (±0.4)	122.3 ^b^ (±0.3)	1926.0 ^a^ (±0.8)	668.6 ^a^ (±0.6)	348.9 ^a^ (±0.2)	35.5 ^b^ (±0.4)	515.2 ^b^ (±0.4)	78.7 ^b^ (±0.3)	112.3 ^b^ (±0.2)
	Yellow	114.9 ^b^ (±0.2)	1193.1 ^b^ (±0.7)	441.9 ^a^ (±0.1)	515.0 ^b^ (±0.3)	151.5 ^a^ (±0.3)	851.9 ^b^ (±0.2)	144.1 ^b^ (±0.2)	220.2 ^b^ (±0.6)	41.8 ^a^ (±0.1)	905.8 ^a^ (±0.2)	183.7 ^a^ (±0.4)	265.1 ^a^ (±0.1)
Protocatechuic acid	Red	4.3 ^a^ (±0.1)	170.2 ^a^ (±0.3)	6.4 ^a^ (±0.1)	36.3 ^a^ (±0.2)	1.5 ^a^ (±0.1)	14.7 ^a^ (±0.2)	7.0 ^a^ (±0.1)	3.3 ^a^ (±0.2)	nq	153.7 ^a^ (±0.6)	11.2 ^a^ (±0.1)	3.2 ^a^ (±0.1)
	Yellow	0.873 ^b^ (±0.1)	14.7 ^b^ (±0.4)	6.6 ^a^ (±0.1)	25.3 ^b^ (±0.6)	1.1 ^b^ (±0.2)	9.2 ^b^ (±0.3)	2.1 ^b^ (±0.5)	2.5 ^b^ (±0.3)	nq	88.2 ^b^ (±0.2)	1.3 ^b^ (±0.2)	3.2 ^a^ (±0.4)
4-Hydroxybenzoic acid	Red	7.1 ^a^ (±0.2)	3.4 ^b^ (±0.4)	130.8 ^a^ (±0.1)	113.6 ^a^ (±0.1)	2.2 ^b^ (±0.3)	2.5 ^b^ (±0.2)	7.2 ^b^ (±0.2)	2.6 ^a^ (±0.1)	nq	5.5 ^a^ (±0.6)	nq	nq
	Yellow	5.4 ^b^ (±0.1)	123.9 ^a^ (±0.4)	12.1 ^b^ (±0.1)	115.0 ^a^ (±0.1)	3.1 ^a^ (±0.5)	40.8 ^a^ (±0.3)	9.7 ^a^ (±0.2)	nq	nq	3.4 ^b^ (±0.2)	4.8 ^a^ (±0.4)	nq
Caffeic acid	Red	nq	8.6 ^a^ (±0.1)	4.3 ^b^ (±0.1)	15.1 ^b^ (±0.2)	nq	19.7 ^a^ (±0.2)	4.3 ^a^ (±0.1)	nq	nq	4.4 ^b^ (±0.3)	nq	nq
	Yellow	nq	6.3 ^b^ (±0.2)	5.0 ^a^ (±0.6)	115.0 ^a^ (±0.4)	nq	12.9 ^b^ (±0.3)	nq	16.1 ^a^ (±0.5)	nq	9.2 ^a^ (±0.1)	nq	nq
Vanillic acid	Red	4.3 ^a^ (±0.1)	11.1 ^a^ (±0.3)	174.5 ^a^ (±0.1)	125.2 ^a^ (±0.2)	nq	122.2 ^a^ (±0.1)	156.1 ^a^ (±0.2)	4.1 ^a^ (±0.2)	3.2 ^b^ (±0.5)	4.8 ^b^ (±0.1)	13.3 ^b^ (±0.6)	23.4 ^a^ (±0.5)
	Yellow	2.8 ^b^ (±0.1)	7.3 ^b^ (±0.1)	10.9 ^b^ (±0.2)	11.9 ^b^ (±0.3)	5.4 ^a^ (±0.6)	121.1 ^a^ (±0.3)	128.5 ^b^ (±0.7)	3.7 ^b^ (±0.1)	5.4 ^a^ (±0.3)	120.6 ^a^ (±0.2)	146.4 ^a^ (±0.4)	12.5 ^b^ (±0.7)
Syringic acid	Red	nq	87.4 ^a^ (±0.1)	18.2 ^a^ (±0.3)	6.0 ^b^ (±0.1)	nq	101.4 ^a^ (±0.6)	13.1 ^a^ (±0.1)	nq	nq	6.9 ^b^ (±0.1)	3.7 ^b^ (±0.1)	6.4 ^a^ (±0.2)
	Yellow	nq	9.7 ^b^ (±0.4)	12.3 ^b^ (±0.4)	4.7 ^a^ (±0.1)	nq	96.4 ^b^ (±0.2)	8.9 ^b^ (±0.7)	nq	nq	89.9 ^a^ (±0.1)	95.2 ^a^ (±0.2)	4.1 ^b^ (±0.4)
*p*-Coumaric acid	Red	8.2 ^a^ (±0.2)	26.9 ^a^ (±0.8)	3.5 ^b^ (±0.8)	5.0 ^b^ (±0.3)	nq	8.0 ^a^ (±0.1)	nq	nq	nq	9.0 ^a^ (±0.3)	nq	2.0 ^b^ (±0.1)
	Yellow	2.6 ^b^ (±0.7)	8.3 ^b^ (±0.2)	3.9 ^a^ (±0.1)	26.7 ^a^ (±0.5)	1.4 ^a^ (±0.1)	3.1 ^b^ (±0.4)	2.6 ^a^ (±0.2)	nq	10.5 ^a^ (±0.1)	7.3 ^b^ (±0.7)	nq	3.6 ^a^ (±0.5)
Ferulic acid	Red	nq	nq	nq	22.6 ^b^ (±0.2)	nq	nq	nq	nq	nq	nq	nq	nq
	Yellow	nq	nq	nq	27.2 ^a^ (±0.3)	nq	5.1 ^a^ (±0.1)	nq	nq	15.0 ^a^ (±0.2)	1.9 ^a^ (±0.5)	nq	3.2 ^a^ (±0.2)
Ellagic Acid	Red	8497.2 ^a^ (±0.3)	nq	nq	nq	nq	nq	nq	nq	987.1 ^a^ (±0.2)	nq	8114.0 ^a^ (±0.2)	nq
	Yellow	nq	nq	nq	793.1 ^a^ (±0.1)	nq	nq	nq	nq	nq	949.9 ^a^ (±0.5)	7565.9 ^b^ (±0.1)	420.1 ^a^ (±0.2)

* F: Free phenolic compounds; E: Esterified phenolic compounds; G: Glycosylated phenolic compounds; I: Insoluble phenolic compounds; nq: below the limit of quantification (LOQ). Means followed by the same letter in the column do not differ, according to the Tukey test (*p* ≤ 0.05).

## Data Availability

The datasets generated for this study are available on request to the corresponding author.

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
