# Peer review of "Exploring Anthocyanin and Free and Bound Phenolic Compounds from Two Morphotypes of Araçá (Psidium cattleianum Sabine) by LC-ESI-QqQ-MS/MS"

_foods, 2023, doi:10.3390/foods12173230_

Round 1

Reviewer 1 Report

Reviewer`s Comments (major revision):

This manuscript had extracted anthocyanin, free and bound phenolic compound contents from two morphotypes of areaca, then quantified them by LC-ESI-QqQ-MS/MS. This research was useful in understanding the nutrition of different morphotypes of araca. However, some critical points need to be addressed. The following are some suggestions and requests and the revised manuscript need to be re-reviewed.

1. The abstract of a research paper should contain a statement of the purpose of the study, methods, data analysis and main conclusion. Please promote the statement of the abstract, especially avoiding introductive description like the first sentence. 2. In the keywords, I believe LC-MS is essential. 3. The content of introduction section is less, and the background is not clearly clarified. Please supplement some introductions to give readers more understanding of related research field. 4. At the bottom of the introduction section, please make a comparison with existing literatures and clarify the innovation of this manuscript. What is your advantage compared with reported works? 5. This manuscript is about extraction and quantification by LC-MS. International research progress of extraction should be supplemented in the introduction section, especially green aqueous two phase extraction with the important works cited (Functionalized aqueous biphasic system coupled with HPLC for Highly sensitive detection of quinolones in milk; Extraction and recovery of phenolic compounds from aqueous solution by thermo-separating magnetic ionic liquid aqueous two-phase system; Selective separation and simultaneous recoveries of amino acids by temperature-sensitive magnetic ionic liquid aqueous biphasic system. 6. In Line 157, the abbreviation (MRM) should be annotated by its full name in the first appearance. Please carefully check throughout the whole manuscript to solve this problem. Also, where did the data of Table 1 come from? If they are from literatures, please cite them in Table 1. If they are obtained from experiments, please give the experimental condition. 7. For better understanding of your work, some reported chromatographic methods should be briefly introduced, such as cellulose based chromatography with the important literatures quoted (Bisphosphonated-immobilized porous cellulose monolith with tentacle grafting by atom transfer radical polymerization for selective adsorption of lysozyme; Preparation of cellulose-based chromatographic medium for biological separation: A review; Application of cellulose to chromatographic media: Cellulose dissolution, and media fabrication and derivatization. 8. This work mainly focused on the extraction of phenolic compounds. So please briefly introduce the background of phenolic compounds extraction, such as ionic liquid and deep eutectic solvent extraction, with these important literatures cited (Determination and correlation of phase equilibria of chiral magnetic ionic liquid aqueous two-phase systems with different inorganic salts at 298.15 K; Liquid-liquid equilibria for (polypropylene glycol 400 based magnetic ionic liquids + inorganic salts) aqueous two-phase systems at 298.15 K; Extractive resolution of racemic phenylalanine and preparation of optically pure product by chiral magnetic ionic liquid aqueous two-phase system. 9. Please provide some important chromatograms for the separations of phenolic compounds and anthocyanins. 10. Figure 3 is not clear, please redraw this figure and give us clear presentation. 11. In the conclusion section, please demonstrate some important data to support your conclusion. Also, the innovations of this manuscript should be summarized.

The quality of English is not bad and need moderate editing.

Reviewer 2 Report

Dear Authors, 

The manuscript entitled "Exploring anthocyanin and free and bound phenolic compound  contents from two morphotypes of araçá (Psidium cattleianum  Sabine) by LC-ESI-QqQ-MS/MS" has interesting topic. However, some improvements are needed before further action. In the introduction, you should write more about the problem statement and how your study is going to solve the current problems. In the conclusion, your conclusive remarks as well as implication of your study are missing. You need to give the readers some take-home messages to understand the benefits of your study. My specific comments are shown in pdf file. 

Thank you

Minor improvements are needed.

Reviewer 3 Report

The comments for authors:

1. Abstract: More key-finding has to be included. Overall conclusion is missing.

2. Introduction: The authors should explain why they decided to examine the composition of this fruit. What is its significance?

3. Subsection 2.2. Moisture content and particle size of the sample material has to be added.

4. Subsection 2.3. Why did the authors not use Food Grade solvents for extraction? Why did the authors not use green and advanced extraction techniques in order to improve the efficiency of the process? Ultrasound power has to be added.

5. Figure 1. Flowchart of the experimental process has to be improved, since it is confusing, it is not clear from the graph which course the experiment went on.

6. Subsection 2.4. The method by Neves et al. has to be described in brief with modifications included. Why five extractions were carried out?

7. Subsection 3.2. Line 236. "Due to the differences between the techniques used, it is...". Which techniques?

8. Table 2. The authors should indicate in Table 2 which compound is identified and quantified for the first time in red and yellow araçá.

9. Conclusions: Based on the obtained results, it is necessary to state their practical application.

Reviewer 4 Report

The publication presented for evaluation was written very carefully. The research methodology was described in detail. In addition, a valuable enrichment is the chemometric analysis. The graphs have been correctly and carefully prepared, but the text should be corrected in terms of punctuation (spaces, L-liter capital letters, etc.).

A comparison of the content of individual compounds between red and yellow araçá would be an interesting addition (statistical analysis).

Round 2

Reviewer 1 Report

There are also some critical issues need to be addressed. The revised manuscript need to be re-reviewed. 

The authors claimed that “unlike the other cited studies, our work separately analyzed phenolic compounds in all parts of fruit, peel, pulp and seed.”. This is not true. Many related literatures have already done this kind of work. So, what is the innovation of this manuscript? Applying common method to do common work? Please carefully rewritten your innovations at the bottom of introduction section. These authors must be aware that “Foods” is a high-level journal. Please take it seriously and carefully correct your manuscript.

In Figure 3, these chromatograms are not suitable at all. All these phenolic compounds were not detected separately. How did authors give several separate chromatograms? All these compounds should be simultaneously demonstrated in one chromatogram with satisfactory base-line separation and good peak shape. Moreover, many peaks of Figure 3 could not satisfy the standard of pure compound. How did authors guarantee all these phenolic compounds were separated? They probably contained large amount of impurity. How did authors define their structures by MS when these compounds were even not pure? Large amount of evidences from MS should be given to me for re-evaluation. This is the most important problem which should be carefully addressed.

The background introduction is obviously insufficient and should be carefully expanded. Many of my previous review suggestions were not addressed. Please seriously refer to them and carefully correct your manuscript.

English need moderate revisions.
